# Fusing Visual and Textual Cues for Sequential Image Difference Captioning

## Abstract

We present Fusing Visual and Textual Cues (FVTC); a novel technique for image difference captioning that is able to benefit from additional visual and/or textual inputs. FVTC is able to succinctly summarize multiple manipulations that were applied to an image in a sequence. Optionally, it can take several intermediate thumbnails of the image editing sequence as input, as well as coarse machine-generated annotations of the individual manipulations. We demonstrate that the presence of intermediate images and/or auxiliary textual information improves the model's captioning performance. To train FVTC, we introduce METS (Multiple Edits and Textual Summaries) – a new open dataset of image editing sequences, with textual machine annotations of each editorial step and human edit summarization captions after the 5th, 10th and 15th manipulation. [1]

## 1 Introduction

With recent advancements in Generative AI, image manipulation becomes increasingly easier to perform and harder to notice, motivating new techniques for auditing the provenance and edits made to an image. Often, multiple edits are applied in sequence by one or multiple editors, forming a provenance graph containing multiple versions of the same image at different stages of the editing process. To avoid the spread of misinformation, it is important to be able to communicate the history of these changes to the end user succinctly to enable them to make informed trust decisions Gregory (2019).

Image difference captioning (IDC) usually aims to generate a difference caption given two images, the original and the edited one, regardless of the number of manipulations applied to the image. In this work, we explore image difference captioning with multiple inputs (IDC-MI), assuming access to multiple snapshots of the image editing sequence and/or auxiliary information about each individual edit. This commonly arises during a creative supply chain where multiple editors contribute to a final image. For example, emerging metadata standards for media provenance, such as the Coalition for Content Provenance and Authenticity (C2PA) Coalition for Content Provenance and Authenticity (2023) collect rich information on this edit process as a provenance graph. This data structure contains multiple versions (thumbnails) of the image at different stages of the editing process, and optionally textual short descriptions of changes made. One use case for IDC-MI is aggregate this multi-modal context and summarize it in a short textual description.

The first challenge in edit sequence captioning is the limited availability of training data. Most datasets for image difference captioning focus on image pairs rather than longer sequences. While the Magic Brush Zhang et al. (2024) dataset does provide multi-turn editing sequences, they are limited to three steps at most. Furthermore, all of the edits are applied to different non-overlapping objects, meaning that the final summary of all the manipulations could be constructed from a concatenation of the description of the individual steps. However, in real scenarios, the edits can be applied to the same area, potentially in a destructive or mutually exclusive manner, and the final summary should only describe the salient, still visible changes. For example, suppose the first manipulation changes the color of a bicycle, and the second one replaces the bicycle with a car. In that case, the final summary should not mention the color change as it is irrelevant to the final result. The second challenge lies in developing a methodology capable of handling interleaved multi-modal in-

---

[1]The METS dataset will be released for open access upon acceptance.

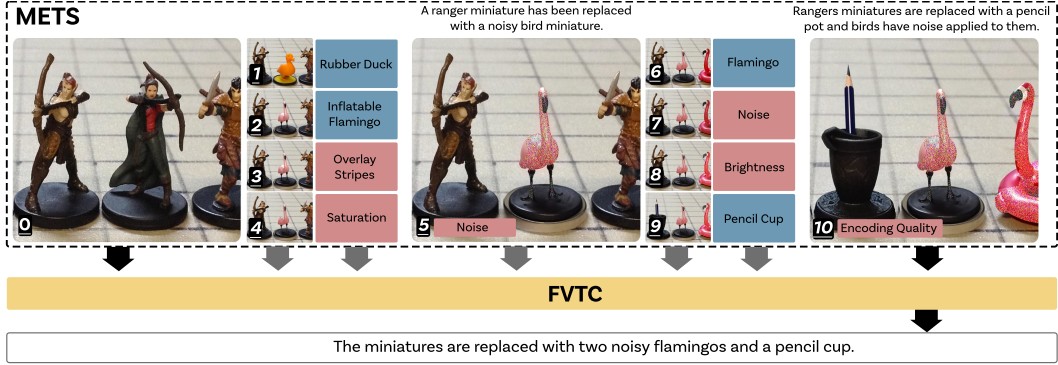

Figure 1: FVTC is capable of processing sequences of images, optionally accompanied by coarse edit annotations, to produce a succinct and informative summary of the differences. We train it with METS – a novel dataset of long image editing sequences paired with machine annotations and human-written summaries at multiple steps. Optional image and text inputs are denoted with gray arrows.

puts. Many existing image difference captioning architectures are designed with exactly two image inputs in mind and would not be able to scale beyond that, either due to architectural constraints or memory limitations. The contributions of this paper are twofold:

1. First, we introduce METS (Multiple Edits and Textual Summaries) – a dataset of image editing sequences, with textual machine annotations of each editorial step and human edit summarization captions after the 5th, 10th, and 15th manipulation.

2. We train FVTC (Fusing Visual and Textual Cues) – a multi-modal LLM with multiple visual inputs and provide a comprehensive evaluation of the benefits of both additional visual and textual inputs.

We demonstrate that the presence of intermediate images and/or auxiliary textual information improves the model's captioning performance. Additionally, we demonstrate that fine-tuning a model trained on other synthetic data with METS helps to bridge the domain gap and improves zero-shot performance on real-life images. The illustration of FVTC and METS is shown in Fig. 1.

## 2  RELATED WORK

Image difference captioning (IDC) is closely related to image captioning and visual question answering, both requiring a visual understanding system to model images and a language understanding system capable of generating syntactically correct captions. The revolution of IDC in recent years depends heavily on the advent of visual and text modeling approaches, together with cross-domain learning techniques that bridge the representation gap between them.

Initial methodologies for modeling visual content involve incorporating overarching CNN features such as VGG Donahue et al. (2015), and ResNet Rennie et al. (2017) into text generation models. This integration capitalizes on the dense and meaningful representations these models provide. To enhance the representation of multiple objects and their interrelations, various techniques have emerged. Some methods Lu et al. (2017); Gu et al. (2018); Anderson et al. (2018); Huang et al. (2019), partition images into discrete patches, extracting CNN features from each. Conversely, certain methodologies opt to utilize the outputs from an early ResNet layer, effectively capturing spatial attributes in a gridded format. In contrast, Cornia et al. (2020); Anderson et al. (2018); Huang et al. (2019) employ Region Proposal Network (RPN) to extract features from potential object candidates, thereby improving alignment with the semantic entities referenced in paired captions. Other avenues of exploration include graph-based Yang et al. (2019) and tree-based networks Yao et al. (2019), aiming to capture object relations across varying levels of granularity.

Traditionally, RNN/LSTM architectures Graves & Graves (2012) have dominated text modeling owing to their intrinsic sequential nature. Variants like single-layer RNN Vinyals et al. (2015);

Mao et al. (2015) or double-layer LSTM Donahue et al. (2015); Anderson et al. (2018); Yao et al. (2019) are commonly utilized, often coupled with diverse methods to embed image features more deeply into the recurrent process, such as additive attention Stefanini et al. (2022). During inference, captions are generated in a step-by-step manner, where the prediction of each word depends on all preceding words. Although this enhances linguistic coherence, RNN/LSTM-based approaches face challenges in modeling lengthy captions. Recent transformer-based methodologies, like those employing a full-attention mechanism Luo et al. (2021); Wang et al. (2021); Cornia et al. (2020), have alleviated this issue. Advanced transformer-based models such as BERT Devlin et al. (2018), GPT Brown et al. (2020), and LLaMA Touvron et al. (2023a) have demonstrated success across diverse visual-language tasks Hu et al. (2022); Mokady et al. (2021); Gao et al. (2023); Zhang et al. (2021); Li et al. (2020).

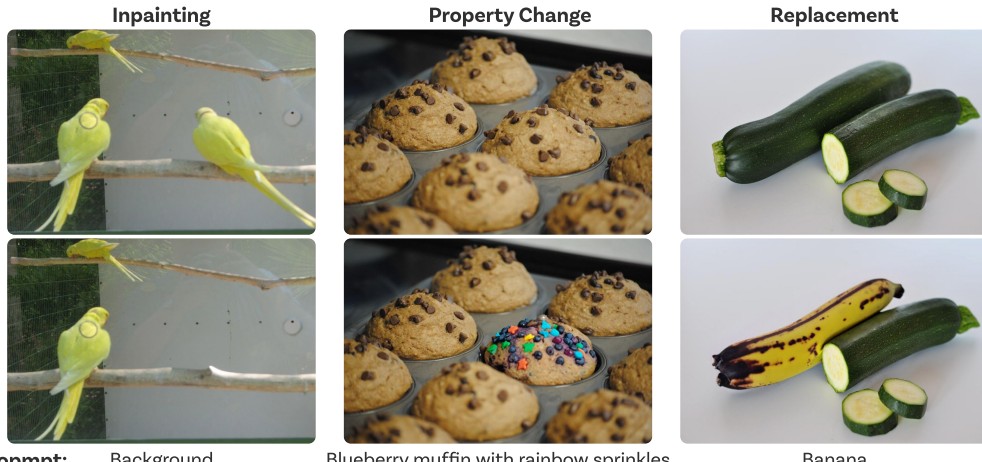

Figure 2: Illustration of the different types of manipulations performed using Firefly Generative Fill. **(left)** Inpainting is done by using the word *background* as the prompt. **(middle)** property change is done by prompting GPT3.5 to output a likely change in color, material, texture or other applicable property of the object. **(right)** replacement is done by prompting GPT3.5 to output a likely replacement candidate object that would be a close match to the shape of the original, but different semantically.

The objective of visual language modeling is to establish connections between image/video and text representations, catering to specific tasks like joint embedding (e.g., CLIP Radford et al. (2021) and LIMoE Mustafa et al. (2022) for cross-domain retrieval), text-to-image tasks (e.g., Stable Diffusion Rombach et al. (2022) for text-based image generation, InstructPix2Pix Brooks et al. (2022) for image editing), and image-to-text tasks (e.g., visual question answering Alayrac et al. (2022); Wang et al. (2021), visual instructions Gao et al. (2023); Driess et al. (2023)). In the realm of image captioning, strategies for mapping images to text can be classified into two main approaches. The first approach involves the early fusion of image and text features to enhance alignment between image objects and textual descriptions Tsimpoukelli et al. (2021); Mokady et al. (2021); Wang et al. (2021); Li et al. (2020). These methods employ BERT-like training strategies, where a pair of images and a masked caption are inputted, replacing the masked words during inference with either a start token or a prefixed phrase like 'A picture of'. The second approach centers on learning a direct conversion from image to text embedding. Initial CNN-based methods incorporate image features as the hidden states of LSTM text modules Donahue et al. (2015); Vinyals et al. (2015); Yao et al. (2019); Karpathy & Fei-Fei (2015); Rennie et al. (2017), whereas later transformer-based techniques favor cross-attention mechanisms Luo et al. (2021); Cornia et al. (2020). Notably, recent trends in both approaches involve harnessing powerful pretrained large language and vision models to establish a straightforward mapping between the two domains Merullo et al. (2022); Eichenberg et al. (2021); Li et al. (2023); Tsimpoukelli et al. (2021); Mokady et al. (2021); Chen et al. (2023).

Image difference captioning represents a specialized form of image captioning, aiming to disregard common objects across images and instead accentuate subtle alterations between them. Pioneering this domain, Spot-the-Diff Jhamtani & Berg-Kirkpatrick (2018) introduces potential change clusters,

employing an LSTM-based network to model them. However, their approach relies on pixel-level differences between input images, rendering it sensitive to noise and geometric transformations. In contrast, DUDA Park et al. (2019) computes image differences at the semantic level using CNNs, enhancing robustness against minor global alterations. Several approaches extend the foundation laid by DUDA. For example, SRDRL+AVS Tu et al. (2021b) initially assesses the correlation between the subtracted difference and image pairs to ascertain the occurrence of the change. Subsequently, it incorporates part-of-speech information to dynamically leverage visual data. M-VAM Shi et al. (2020) and VACC Kim et al. (2021) propose a viewpoint encoder to mitigate viewpoint disparities, while VARD Tu et al. (2023a) suggests a viewpoint invariant representation network to explicitly capture changes. Additionally, Sun et al. (2022) integrates bidirectional encoding to refine change localization, and NCT Tu et al. (2023b) utilizes a transformer to aggregate neighboring features. These methodologies concentrate on the image modality, exploiting benchmark-specific characteristics such as nearly identical views in Spot-the-Diff Jhamtani & Berg-Kirkpatrick (2018) or synthetic scenes with limited objects and change types in CLEVR Park et al. (2019). More recently, IDC-PCL Yao et al. (2022) and CLIP4IDC Guo et al. (2022) have adopted BERT-like training approaches to model difference captioning language, achieving state-of-the-art performance.

## 3 METHODOLOGY

In this section, we describe the methodology behind the dataset generation as well as IDC model training. Subsection 3.1 describes the data generation process used to create the METS dataset. Subsection 3.2 describes the architecture of the model used to train on the METS dataset to perform the multi-input image difference captioning task.

### 3.1 DATA GENERATION

We generate a dataset of image editing sequences, with textual machine annotations of each editorial step and human edit summarization captions after the 5th, 10th, and 15th manipulation, as shown in Fig. 4. Binary masks of the manipulation regions at each step are also included. Our dataset covers a wide variety of pixel-level and generative manipulations. The prompt for each manipulation is generated using GPT-3.5 to ensure plausible and diverse manipulations.

### 3.1.1 INDIVIDUAL EDITS

We identify two main categories of edits: pixel-level and generative manipulations. Pixel-level edits are simple manipulations such as changing the brightness of an image or applying a blur filter. Generative manipulations change the semantic content of the image, altering the story that the image tells.

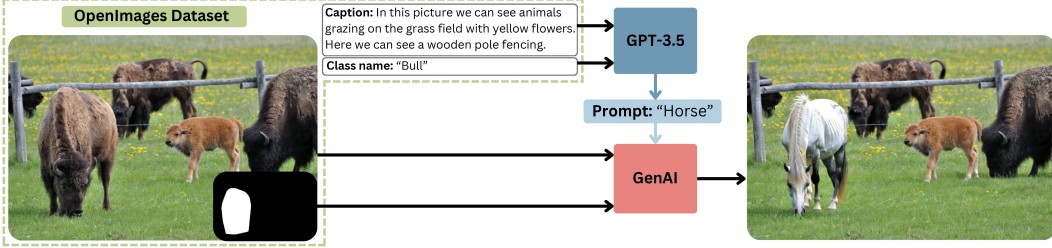

Figure 3: METS image generation pipeline for generative manipulations. The image, its localized narrative, object class name, and segmentation mask are sampled from the OpenImages dataset. The localized narrative and class name are used to construct a prompt for GPT3.5, which outputs a likely replacement candidate object or a property change. The prompt templates are manipulation-type specific and can be seen in suppmat. In the case of inpainting, the GPT3.5 block is omitted, and the prompt is simply *background*. The pre-processing of the segmentation mask ensures that no part of the object remains outside of the mask. It involves generating a convex hull of the mask and applying dilation to it. The generative manipulation is then conditioned on the image, the mask, and the prompt and applied using Firefly Generative Fill.

Pixel level manipulations are performed using the AuglyPapakipos & Bitton (2022) image augmentation library, with a random choice of augmentation type and parameters. Augmentation types include brightness, contrast, saturation, and encoding quality changes; blur, noise and sharpness filters; and overlaying random stripes of the color of different widths.

We further divide generative manipulations into three categories: **inpainting** where an object is removed from the image, **replacement** where an object is replaced with another object, and **property change** where the object's material properties are altered. We illustrate different types of manipulations in Fig. 2.

Generative manipulations are applied using Firefly Generative Fill[2], which is a language-guided inpainting model. In addition to the image itself, the model is provided with a segmentation mask and a text prompt. We generate a convex hull of the segmentation mask and apply dilation to it to ensure that no part of the object remains outside of the mask. The origin of the text prompt depends on the type of manipulation. For **inpainting** we use the word *background*, which was shown to perform on par with inpainting-specific models. For **replacement**, we further illustrate the image editing pipeline in Fig. 3, where we use GPT3.5 in a few-shot learning manner, prompting with a localized narrative for the whole image, a bounding box of the mask, and the class label of the mask to come up with a probable replacement candidate object that would be a close match to the shape of the original object. We use a similar strategy for **property change**, but prompting GPT3.5 to output a likely property change.

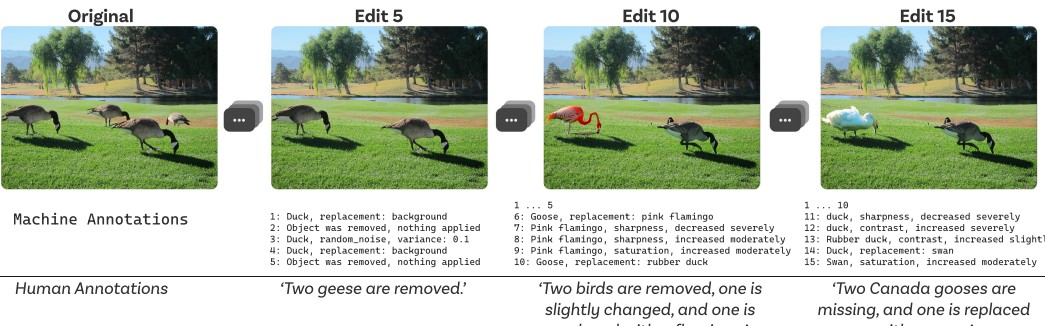

Figure 4: An example of a sequence of manipulations in METS. The original image is shown in the first column, followed by the manipulated images. The binary masks of the manipulated regions are superimposed on the images. The machine annotations generated during the sequence creation are shown in orange, while the human annotations are shown in blue. Note that only edit steps 5, 10, and 15 are shown, as these are the steps for which human annotations were collected. All other data types are available for all steps.

### 3.1.2 SEQUENCE GENERATION

We sample the images from the OpenImages dataset, making use of the provided segmentation masks and localized narratives. We choose the images with at least 5 non-overlapping segmentation masks. We then follow a procedure illustrated in Fig. 5 to apply a sequence of edits to the image. At each iteration step, we pick a segmentation mask and either apply a generative or a pixel-level manipulation to that area of the image or move on to the next mask. The probability of switching to the next mask is proportional to the number of manipulations already applied to the mask.

Formally, we define the probabilities of applying a generative manipulation $P_g$, a pixel-level manipulation $P_p$ and moving on to the next mask $P_n$ as follows:

$$P_g = g - \frac{n}{2}, \quad P_p = (1 - g) - \frac{n}{2}, \quad P_n = 1 - P_g - P_p, \tag{1}$$

where $g = 0.9$ if no generative manipulations have been applied to the mask yet and $g = 0.1$ otherwise. The value of $n$ is proportional to the number of manipulations already applied to the

---

[2]https://firefly.adobe.com/upload/inpaint

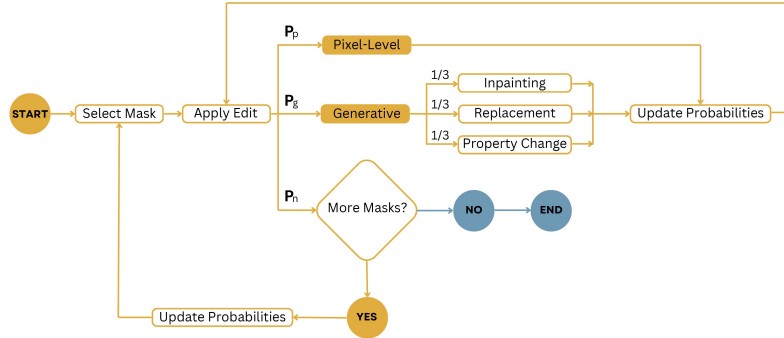

Figure 5: The diagram of the sequence generation process. For each image, we first go through up to 15 segmentation masks and apply edits, chosen randomly, where the probabilities of choices depend on the number of edits already applied to the mask. The probability of applying a generative manipulation is greatly lowered if a generative manipulation has already been applied. This lowers, but does not eliminate, the chance of making destructive or mutually exclusive manipulations.

mask, defined as follows:

$$n = \max(0, \frac{40 \times (i - i_{min})}{100}), \tag{2}$$

where $i$ is the current step and $i_{min}$ is the minimum number of steps required to move on to the next mask. We set $i_{min} = 5$.

After each manipulation step, we record the type of manipulation, the parameters of the manipulation, and the binary mask used to apply the manipulation. This information is saved in a text format. For pixel-level manipulations, the text format is as follows:

Object: obj_name, manipulation: edit_name, intensity: intensity

where `obj_name` is the name of the object as annotated within the OpenImages dataset, `edit_name` is the manipulation type and `intensity` is chosen at random from a set of predefined parameters, individual for each manipulation type.

For generative manipulations, the text format is as follows:

Object: obj_name, replacement: prompt

where `prompt` is either *background* for inpainting or the output of GPT3.5 for replacement and property change manipulations. Examples of the template-generated text can be seen from Fig. 4, marked as *machine annotation*.

As a result, for each input image, we obtain a sequence of manipulated versions applied on top of each other and a list of annotations describing each manipulation step type, parameters, and location. We generate 1000 such sequences with an average of 21.4 steps per sequence.

### 3.1.3 LABELLING

We collect human annotations for difference summarization at the 5th, 10th, and 15th step of the manipulation sequence. In each task, the users are presented with the input image $I$ and an output image $I'_n, n \in [5, 10, 15]$ and are asked to provide a short one-sentence summary of all of the differences they see between the two images. Examples of such summaries can be seen in Fig. 4, marked as *human annotations*.

### 3.2 ARCHITECTURE

Our architecture is illustrated in Fig. 6. Our setup consists of a Vision Transformer (ViT) Dosovitskiy et al. (2021) image encoder and the open-sourced LLaMA2-chat (7B) large language model Touvron et al. (2023b). The visual tokens are concatenated in groups of 4 and projected to the language

Figure 6: Architecture diagram of the model. The LLaMA-2 language model is conditioned using the multi-modal instruction template, which includes at least two image features and optional auxiliary textual information. All optional content is placed within dashed boxes. The image features extracted from the ViT image encoder are concatenated in groups of 4 and projected to the LLM embedding space with a linear projection layer. The visual encoder weights are frozen, and only the language model and the projection layer are trained.

model's embedding space with a linear projection layer. During training, the visual encoder weights are frozen, and only the language model and the projection layer are trained.

Uniquely, we use multiple images as input to the model and train it for the task of image difference captioning. We note that this approach is capable of handling an arbitrary number of input images, which allows us to input several snapshots of the image editing sequence at once.

Optionally, we provide the model with auxiliary textual information in the form of machine annotations, described in Section 3.1. The annotations for each manipulation are interleaved with the image features and are used to guide the model's attention to the relevant parts of the image.

We follow the multi-modal instructional template from Chen et al. (2023) and adjust it to our task:

$$[INST] <ImageFeature></Img> T \ldots <ImageFeature></Img> T [idc]$$
$$ins [/INST]$$

where the image feature tags are repeated for each input image in the sequence, $T$ is the optional auxiliary textual information, `[idc]` is the task identifier for image difference captioning and `ins` is the instruction that is chosen at random from a set of predefined instructions, all synonymous with *describe the defferences between the images.*.

The model is trained to minimize the captioning loss, which is defined as

$$\mathcal{L} = -\sum_{i=1}^{m} l(s^v, s_1^t, \ldots, s_i^t), \tag{3}$$

where $m$ is a variable token length and $l$ is next-token log-probability conditioned on the previous sequence elements

$$l(s^v, s_1^t, \ldots, s_i^t) = \log p(t_i|x, t_1, \ldots, t_{i-1}). \tag{4}$$

### 3.2.1 TRAINING

All of the models are trained on a single A100 GPU with 80GB of memory for 300 epochs with 1000 steps per epoch and a batch size of 6. We use AdamW optimizer with a cosine learning rate

scheduler with an initial learning rate of $10^{-5}$ and a warmup learning rate of $10^{-6}$ for a warmup period of 1000 steps. The input image size is $448 \times 448$, and the maximum token length is 1024.

## 4 EXPERIMENTS

### 4.1 DATASETS

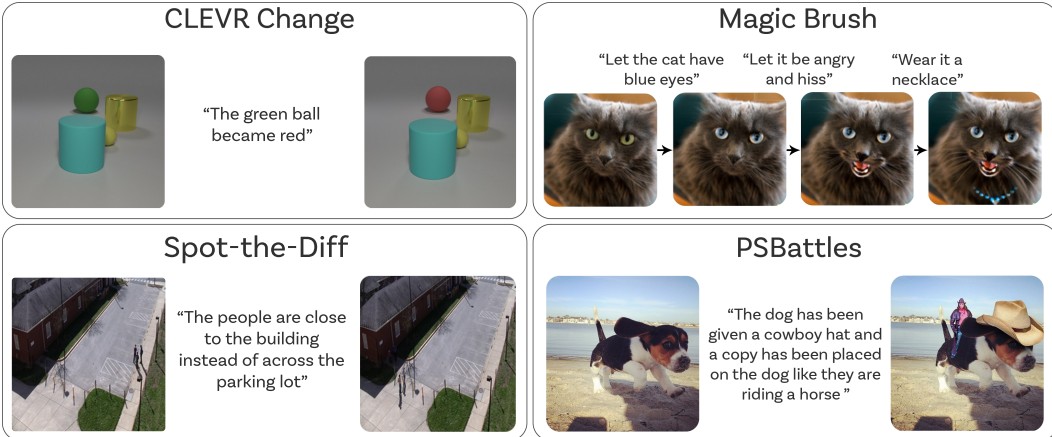

Figure 7: Examples of images and annotations from the CLEVR-Change, Spot-the-Diff, MagicBrush and PSBattles datasets.

In addition to our own dataset, we train and evaluate our model on a number of other datasets used in the image difference captioning literature illustrated in Fig. 7.

**CLEVR-Change** Johnson et al. (2017) consists of 67,660, 3,976, 7,970 training, validation, and test image pairs, respectively. The images are generated using the CLEVR engine and contain renders of primitive 3D shapes. The types of edits include changes in shape, color, material, size, and position of the objects. This dataset serves as a good benchmark due to its large volume and precise annotations. However, the synthetic nature of the images creates a large domain gap, making it difficult to generalize to real-world images.

**Spot-the-Diff** Jhamtani & Berg-Kirkpatrick (2018) is a dataset of 13,192 well-aligned image pairs from CCTV cameras. There are no viewpoint changes, and the edits are limited to object addition, deletion, or movement. The dataset is split into training, validation, and test sets following the official split of 80%, 10% and 10%.

**PSBattles** Heller et al. (2018) is a dataset of real-world image pairs collected from the Reddit Photoshop Battles subreddit. The difference captions for a subset of the dataset were collected by Black et al. (2024) in a user study. We use this dataset for the evaluation of the model's generalization capability to real-world images.

**InstructPix2Pix** Brooks et al. (2022) is a dataset of ∼1M image pairs generated with prompt-to-prompt Hertz et al. (2022) approach. The difference captions are later generated by Black et al. (2024) using chatGPT-3. We use this dataset for pre-training of the model during the evaluation in the PSBattles dataset to assess the benefits of fine-tuning on the METS dataset for domain adaptation.

**MagicBrush** Zhang et al. (2024) contains sequences of edited images generated in a manner similar to ours, but with human supervision. Due to the need for human supervision, the maximum length of the sequences is limited to 3 steps. Of 878 training sequences, only 304 have a length of 4 (including the original image), and 547 have a length of 3. We use this dataset to evaluate the model's performance in the IDC-MI setting, using only the samples that have a length of 4. The target annotation is a concatenation of the instructions for each step. As input, we use either the first and the last image in the sequence or all four images in the sequence.

## 4.2 Evaluation

We evaluate the performance of our model in two different settings: standard IDC with two images as input and 'image difference captioning with multiple inputs' (IDC-MI). The former setting is the most common in the literature, while the latter is a novel setting that we introduce in this work.

We evaluate the performance of our model on the standard IDC setting on the CLEVR-Change, InstructPix2Pix, and PSBattles datasets. We evaluate the performance of our model in the IDC-MI setting on the MagicBrush and our proposed METS datasets. In both cases, we use the standard n-gram based metrics BLEU-4 (B4), CIDEr (C), METEOR (M), ROUGE-L (R) and SPICE (S) to evaluate the performance of our model. Additionally, we use LLM-as-judge metric to assess the semantic similarity of the captions that n-gram based metrics struggle to capture. We use GPT4 to score the semantic similarity of each text pair as 'low', 'medium' or 'high' and report the percentage of medium and high scores.

### 4.2.1 Evaluating IDC with Multiple Inputs

Table 1: Performance evaluation in the IDC-MI setting shows BLEU-4 (**B4**), CIDEr (**C**), METEOR (**M**), ROUGE-L (**R**) and LLM as judge medium (**L (M)**) and high (**L (H)**) scores. We report the performance of our model and compare it with GPT3.5 and GPT4-V, varying the number of input images and the presence of auxiliary textual information.

| model | images | text | B4 | C | M | R | L (M) | L (H) |
|---|---|---|---|---|---|---|---|---|
| METS | | | | | | | | |
| GPT3.5 Brown et al. (2020) | 0 | yes | 1.6 | 8.6 | 10.4 | 15.1 | 16.2 | 0.6 |
| GPT4-V *et al.* (2024) | 2 | no | 4.0 | 18.6 | **14.0** | 20.3 | 22.2 | 2.6 |
| GPT4-V*et al.* (2024) | 2 | yes | 1.3 | 0.3 | 11.5 | 13.5 | 19.7 | 0.9 |
| GPT4-V*et al.* (2024) | 4 | no | 3.0 | 15.1 | 13.4 | 19.9 | 26.9 | 1.9 |
| GPT4-V*et al.* (2024) | 4 | yes | 1.4 | 0.4 | 11.6 | 12.9 | 24.1 | 1.2 |
| FVTC-2 (ours) | 2 | no | 5.8 | 20.7 | 11.4 | 23.1 | 22.6 | 9.4 |
| FVTC-2T (ours) | 2 | yes | 7.8 | 25.8 | 13.0 | 26.0 | 24.3 | 11.0 |
| FVTC-4 (ours) | 4 | no | 6.6 | 23.5 | 12.3 | 24.3 | 22.6 | 9.6 |
| FVTC-4T (ours) | 4 | yes | **8.2** | **25.9** | 13.4 | **26.3** | **30.1** | **12.4** |
| MagicBrush | | | | | | | | |
| FVTC-2 (ours) | 2 | no | 4.9 | 29.4 | 13.3 | 28.1 | - | - |
| FVTC-4 (ours) | 4 | no | **6.8** | **44.5** | **15.6** | **31.2** | - | - |

For the IDC-MI setting, we evaluate the model's performance while varying the number of input images and the presence of auxiliary textual information. The intermediate images are sampled to be equally spaced in the sequence, and the textual information is provided in the form of machine annotations described in Section 3.1. We compare the performance of our model with GPT4-V, which has multi-modal capabilities and is capable of taking multiple images and/or text as input. Additionally, we compare with GPT3.5, which serves as a text-only baseline, taking as input only the auxiliary text and no images.

The results of the IDC-MI setting are shown in Table 1. We demonstrate that our method is able to take advantage of the additional inputs, achieving the best performance when both intermediate images and auxiliary textual information are present. On the other hand, GPT4-V suffers from the addition of intermediate images and text, showing a decrease in performance in both cases.

Compared to the base case of just two-image input, the addition of text to our model improves the performance by an average of 18.9% across all metrics, and intermediate images improve the performance by an average of 10.1% across all metrics. The combination of both intermediate images and textual information shows an average improvement of 22.4% across all metrics.

On the other hand, the performance of GPT4-V suffers from the addition of intermediate images, decreasing in performance with the addition of both extra images and text.

Table 2: Image difference captioning performance evaluation on the CLEVR-Change and PSBattles datasets. We report the performance of our model and compare it with the state-of-the-art models and report BLEU-4 (**B4**), CIDEr (**C**), METEOR (**M**) and ROUGE-L (**R**) scores.

| MODEL | TRAINING DATA | B4 | C | M | R | S |
|---|---|---|---|---|---|---|
| CLEVR CHANGE | | | | | | |
| DUDA PARK ET AL. (2019) | CLEVR | 47.3 | 112.3 | 33.9 | - | - |
| IFDC HUANG ET AL. (2022) | CLEVR | 49.2 | 118.7 | 32.5 | 69.1 | - |
| $R^3$NET+SSP TU ET AL. (2021A) | CLEVR | 54.7 | 123.0 | 39.8 | 73.1 | - |
| SGCC OLUWASANMI ET AL. (2019) | CLEVR | 51.1 | 121.8 | 40.6 | 73.9 | - |
| NCT TU ET AL. (2023B) | CLEVR | 55.1 | 124.1 | 40.2 | 73.8 | - |
| SRDL+AVS TU ET AL. (2021B) | CLEVR | 54.9 | 122.2 | 40.2 | 73.3 | - |
| VARD TU ET AL. (2023A) | CLEVR | **55.2** | 124.1 | **40.8** | 74.1 | - |
| FVTC-2 (OURS) | CLEVR | 54.7 | **151.8** | 40.0 | **77.1** | - |
| SPOT-THE-DIFF | | | | | | |
| SRDL+AVS TU ET AL. (2021B) | SPOT-DIFF | - | 35.3 | 13.0 | 31.0 | 18.0 |
| $R^3$NET+SSP TU ET AL. (2021A) | SPOT-DIFF | - | 36.6 | 13.1 | **32.6** | 18.8 |
| VARD-LSTM TU ET AL. (2023A) | SPOT-DIFF | - | 39.3 | 13.1 | 33.1 | 17.5 |
| VARD-TRANSFORMER TU ET AL. (2023A) | SPOT-DIFF | - | 30.3 | 12.5 | 29.3 | 17.3 |
| FVTC-2 (OURS) | SPOT-DIFF | - | **45.5** | **13.7** | 28.7 | **19.3** |
| PSBATTLES | | | | | | |
| VIXEN-C BLACK ET AL. (2024) | IP2P | 4.5 | 7.7 | 9.5 | 20.5 | - |
| FVTC-2 (OURS) | IP2P | 5.3 | 10.3 | 10.8 | 22. | - |
| FVTC-2 (OURS) | IP2P + METS | **5.5** | **14.2** | **11.2** | **22.6** | - |

### 4.2.2 EVALUATING IDC WITH TWO INPUTS

We observe that in the IDC setting, shown in Table 2, the model achieves competitive performance on the CLEVR-Change dataset, outperforming the previous state-of-the-art model VARD on the CIDEr and ROUGE-L metrics. On the InstructPix2Pix dataset, the model outperforms VIXEN only on the METEOR metric. However, it shows a better capability to generalize to real-world images, outperforming VIXEN on the PSBattles dataset for all metrics. Additionally, fine-tuning the model on the METS dataset further improves its performance on PSBattles, showing the dataset's ability to bridge the domain gap between synthetic and real-world images.

### 4.3 LIMITATIONS

As with most LLMs, FVTC can occasionally hallucinate details that are not present in the input images. When errors are made, they are most commonly occurrences of miscounting. For example, the model may state that multiple occurrences of an object have been replaced with another instead of a single occurence or vice versa (i.e. use of plural versus singular). The remaining category of failure cases observed involves cases where level of detail may be too succinct, for example stating an object is replaced but not stating with what.

## 5 CONCLUSION

We have introduced a novel task of image difference captioning with multiple inputs and demonstrated that the presence of additional visual and/or textual inputs improves the model's captioning performance. We have introduced METS – a new dataset of long image editing sequences paired with machine annotations and human edit summarization captions. We have trained a multi-modal LLM with multiple visual inputs and provided a comprehensive evaluation of the benefits of both additional visual and textual inputs. Additionally, we have demonstrated that fine-tuning a model that is trained on other synthetic data with METS helps to bridge the domain gap and improves zero-shot performance on real-life images.

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
