# OpenReview forum: "Fusing Visual and Textual Cues for Sequential Image Difference Captioning"
_ICLR.cc/2025/Conference — Submitted to ICLR 2025_

### Official Review · Reviewer_Luxv · 2024-10-19

**Soundness:** 3
**Presentation:** 4
**Contribution:** 4
**Rating:** 8
**Confidence:** 3

**Summary:**

This paper presents Fusing Visual and Textual Cues (FVTC), a novel technique for image difference captioning that summarizes multiple manipulations applied to an image in sequence.  FVTC leverages intermediate image thumbnails and machine-generated annotations of individual edits, improving captioning performance.  To support FVTC, the authors also introduce METS (Multiple Edits and Textual Summaries), a new dataset of image editing sequences with machine annotations for each edit and human summaries at specific points.  This addresses the limitations in existing datasets, which often focus on image pairs or short, non-destructive editing sequences.

**Strengths:**

The paper introduces a novel task: image difference captioning with multiple inputs, including intermediate editing stages and textual annotations.  This expands the scope of existing image difference captioning and addresses real-world scenarios in creative editing workflows. The METS dataset, with its focus on longer editing sequences and destructive changes, is a significant contribution to the field.

The data generation process for METS appears robust, incorporating generative and pixel-level edits with GPT-3.5 generated prompts and human labeling for edit summaries.  The proposed FVTC model, a multi-modal LLM with multiple visual inputs, is well-suited to the task.

Addressing the challenge of summarizing multi-step image edits, particularly those with destructive changes, is a significant step for image provenance and content authenticity verification. The METS dataset and FVTC model offer valuable resources for research in this area.

**Weaknesses:**

The experimental evaluation could be strengthened by including comparisons to a wider range of models, including closed models like Claude and Gemini, as well as open-source models capable of handling multiple images, such as certain LLaVA versions.  The lack of comparison to other models in the MagicBrush evaluation (Table 1) is a notable omission.

**Questions:**

Could you elaborate on the choice of 1000 sequences for the METS dataset? Is there a risk that this dataset size is insufficient for training robust models?

Why weren't other models, such as those used in the METS experiments (GPT3.5, GPT4-V), included in the MagicBrush comparison (Table 1)?

---

> ### Author Response · Authors · 2024-11-27
>
> We thank the reviewer for their feedback and the recognition of METS as a significant contribution to the field.
> We address the reviewer's questions as follows:
>
> ## Questions
>
> 1. **Choice of 1000 sequences for the METS dataset** The number of sequences in the METS dataset is limited by the labelling cost of the human annotations. We are considering releasing a larger sythetic-only
> version without human annotations in the future.
>
> 2. **Why weren't other models included in the MagicBrush comparison** Since the inference of GPT models incurs a cost, we focused on evaluating the proposed dataset first and foremost.

---

### Official Review · Reviewer_M63D · 2024-10-24

**Soundness:** 2
**Presentation:** 2
**Contribution:** 2
**Rating:** 3
**Confidence:** 4

**Summary:**

The paper creates a new dataset for image difference captioning with multi-step editing, which contains more steps than existing benchmarks, by automatically generating difference captions for each step and adding human annotations every 5 editing steps. Additionally, the authors train a multimodal model on the edited image sequence.

**Strengths:**

The idea of leveraging multiple images from intermediate editing stages to generate difference captions and assessing whether this approach outperforms using only the original and final images is intriguing. Furthermore, the paper demonstrates that the introduced METS dataset may be valuable for adapting models to datasets like PSBattles.

**Weaknesses:**

There are several areas in the current version of the paper that could benefit from further improvement.

**Motivation**

The scenario presented in the paper, which assumes access to both the original and the final edited images along with all intermediate editing steps, seems less applicable to real-world situations where only the original and final images are typically available. Relying on intermediate editing steps—and even more so on the auxiliary text that describes each edit—feels more like a limitation than a strength in practical applications.

**Evaluations**

In Table 1, the authors compare their method to zero-shot GPT-4V but limit the evaluation to their new METS dataset without comparing it to MagicBrush, which is only used for comparing FVTC-2 and FVTC-4. Additionally, the use of n-gram-based metrics, which are highly sensitive to caption phrasing and length, may skew the results. For instance, if GPT-4V accurately captures the edits but generates a longer caption, it would score poorly on n-gram similarity. These metrics are mostly reflective of quality when the models compared are trained on the same datasets, and so have similar phrasing. Incorporating an LLM-as-a-Judge evaluation would improve the reliability and robustness of the results. It would be great if the authors could provide a comparison on MagicBrush, as welll as evaluation with LLM-as-a-Judge.

Moreover, in Table 2, the authors compare their model against several older baselines, all of which use weaker backbones compared to FVTC, which is built on LLaMA-2. Since FVTC adopts a standard LLaVA-like architecture [1] and follows conventional training procedures, the contribution of the proposed model appears limited. Moreover, Table 2 does not include an evaluation on MagicBrush, which is the standard benchmark for this task. This omission limits the comparability and relevance of the results. Adding a comparison on MagicBrush would greatly help.

[1] Liu et al. Visual Instruction Tuning. NeurIPS 2023

**Questions:**

How was text-only GPT-3.5 prompted in the Table 1 experiments? Its poor performance is surprising, given that it had access to all editing steps and only needed to summarize them. Providing a few-shot example in the prompt would likely lead to a significant improvement in its results, could the authors clarify whether this was considered or tested, and if not conduct such experiment?

---

> ### Author Response · Authors · 2024-11-27
>
> We thank the reviewer for recognizing the potential value of both the proposed approach and the METS dataset, while also
> providing valuable critique and suggestions for improvement.
>
> We address the reviewer's concerns below:
>
> ## Weaknesses
>
> ### Unrealistic Motivation
>
> The scenario we are considering is motivated by the real-world task of summarizing provenance metadata, to summarize the change history of an image.  Such metadata (C2PA) is adopted across much of the tech industry and making tools to better comprehend this metadata is a real-world challenge.  Such metadata includes the original and final editted image with intermediate image and text data.  Please refer to e.g. [C2PA Manifests](https://opensource.contentauthenticity.org/docs/manifest/manifest-examples).
>
> The other scenario in which both the intermediate editing steps and their auxiliary metadata could be available is in the context of creative workflows,
> where the editing process is recorded and the metadata is generated automatically by the editing software, such as history states in Photoshop.
>
> ### Evaluation
>
> We have included an LLM based evaluation of the methods in Table 1. Please refer to the comment above for details: https://openreview.net/forum?id=rZxwa8JkJW&noteId=ZDpzdoct63
>
> ## Questions
> All of the models are prompted in the save few-shot learning fashion with two examples, below is the full prompt:
>
> ```
> Given a list of manipulations, summarize the manipulations applied to make the last image from the first. Keep your answer to a few short sentences.Intermediate steps should not be added to the answer if they are not visible in the last frame \n
>             Example1: List: Object: Roller skates, replacement: background
>             Object: Roller skates, replacement: skateboards with LED lights
>             Object: Skateboard, replacement: background
>             Object: Skateboard, manipulapion: encoding_quality, intensity: decreased slightly
>             Object: Skateboard, replacement: rollerblades
>             Object: rollerblades, manipulapion: overlay_stripes, intensity: line width: 0.37617131348743227, line color: (170, 233, 137), line angle: 178, line density: 0.6794338640003309, line type: dashed, line opacity: 0.62
>             Object: rollerblades, manipulapion: brightness, intensity: increased moderately
>             Object: rollerblades, manipulapion: encoding_quality, intensity: decreased moderately
>             Object: Roller skates, replacement: background
>             Object: Skateboard, replacement: rollerblades
>             Object: rollerblades, manipulapion: saturation, intensity: decreased moderately\n
>             Summary: Shoes are now noisy; roller skates are replaced with a hover board and a patch.\n
>             Example2: List: Object: Cheese, manipulapion: brightness, intensity: decreased severely
>             Object: Cheese, manipulapion: contrast, intensity: decreased slightly
>             Object: Cheese, replacement: pickle slice
>             Object: pickle slice, manipulapion: sharpness, intensity: increased moderately
>             Object: pickle slice, manipulapion: encoding_quality, intensity: decreased moderately
>             Object: pickle slice, manipulapion: saturation, intensity: decreased severely
>             Object: pickle slice, manipulapion: overlay_stripes, intensity: line width: 0.4923714860638555, line color: (194, 157, 0), line angle: -85, line density: 0.08991256536478587, line type: solid, line opacity: 0.73
>             Object: Cheese, manipulapion: contrast, intensity: increased moderately
>             Object: Cheese, replacement: spicy cheese
>             Object: Cheese, replacement: sushi
>             Object: Cheese, replacement: crackers
>             Object: Cheese, replacement: background
>             Object: Cheese, replacement: crackers
>             Object: crackers, replacement: crackers
>             Object: global, manipulapion: saturation, intensity: increased severely\n
>             Summary: The cheese slices in the risotto bowls are replaced with a sunglass, chocolate pastry, an orange slice, a partially noisy pine cone, and a tennis ball.
>             Your prompt:
>             List: {lst}
>             Summary:
> ```

---

> ### Comment · Reviewer_M63D · 2024-11-27
>
> I appreciate the authors' efforts in conducting additional experiments and providing the few-shot prompts. I highly encourage the inclusion of all prompts used, as well as additional qualitative examples, in the manuscript or appendix. However, I still have a few major concerns that I outline below:
>
> **Motivation**
>
> The task of image difference captioning with multiple images (and text) (IDC-MI) is a sub-problem of difference image captioning (IDC). As such, it is narrower in scope and applies to very specific scenarios. While I agree with the authors that this task may arise in real-life contexts, this does not inherently make it particularly compelling from a research perspective—IDC-MI appears to be a straightforward extension of IDC. That said, the task may be of interest to some researchers and could inspire further work based on the proposed METS dataset.
>
> To my understanding, the main contribution of this paper lies in the introduction of the METS dataset, while the FVTC model represents a standard large multimodal model fine-tuned on this dataset. A more impactful contribution, in my opinion, would involve an extensive analysis of typical failure cases that models encounter for particular editing operations or entity types. For example, testing multiple models on the METS dataset after fine-tuning them could create an interesting benchmark for evaluating model capabilities. In the current version of the paper, the authors fine-tune only their own model on the dataset, which limits comparability since other models are evaluated in a zero-shot setting.
>
> **Additional Concerns**
>
> **Prompt Design.**
> Examining the provided prompts, I now understand why text-only models may struggle. When multiple instances of the same object are present, the auxiliary text does not specify which instances are being edited. For models like GPT-4V (or similar), it would be much more effective to include the images in the few-shot prompts. Otherwise, the text alone might lead to confusion. This might also explain why adding the auxiliary text, intended to provide additional context about the editing steps and simplify the task, results in performance degradation for GPT-4V.
>
> **LLM-as-a-Judge.**
> Thank you for incorporating the LLM-as-a-Judge experiment. While it partially addressed my concerns, my original suggestion (and I apologize if it was not clear) was to use the LLM to assess whether the generated caption *accurately identifies the editing steps* based on the ground-truth caption, not merely to determine if the generated caption *is similar* to the ground truth. This distinction is critical, as a generated caption could differ significantly in style, word choice, or length (causing the LLM to judge it as different from the original caption) while still correctly capturing all the edits made. Incorporating such a comparison would greatly enhance the clarity and interpretability of the results.
>
> Due to all these reasons, I maintain my original score. However, I encourage the authors to carefully consider the feedback provided here and in other reviews, as I believe it offers insights that could significantly improve their paper.

---

> ### Author Response · Authors · 2024-11-27
>
> Thanks for the reply. We are happy to know that you found the additional experiments and the prompt examples useful.  We confirm are happy to include all of this requested data.
>
> Regarding the motivation.  It may be helpful to further clarify the use context for this paper, which helps motivate the dual contributions of both the algorithm and the dataset and why this 'IDC' problem variant is interesting.
>
> There is an emerging standard for fighting fake news called C2PA (see paper for citation).  It describes a data structure that lives inside an image metadata, that describes how an image was made (e.g. what was done – text information, and previous version of the image – visual information).  In many cases this edit history comprises a sequence of visual versions of an image, interleaved with textual information describing the image (also noting that either of visual and textual information may be absent at each step).
>
> The standard is widely adopted by industry and there are many millions if not billions of such images in circulation.  However it is hard for users to understand this complex provenance information.  A textual summary of it would be very helpful .
>
> Thus a technique is required to intuitively summarize the multimodal edit history, which uniquely presents as sequence of visual and textual change descriptions.  Thus we present FVTC and the METS dataset to train/evaluate it.

---

### Official Review · Reviewer_ccBm · 2024-10-30

**Soundness:** 2
**Presentation:** 3
**Contribution:** 2
**Rating:** 5
**Confidence:** 4

**Summary:**

This paper focuses on the task of image difference captioning. It first introduces a dataset, METS, which contains image sequences generated by pixel-level edits and generative manipulations using inpainting models. Then, this paper trained a Fusing Visual and Textual Cues, which can take multiple images and additional textual information when generating captions. The proposed model is compared with baselines on several image difference captioning tasks.

**Strengths:**

The proposed synthetic image sequence generation pipeline is technically sound. The paper is easy to follow.

**Weaknesses:**

1. **Evaluation Metrics.** This is the main concern of this paper, which makes it hard to evaluate the effectiveness of the proposed method. The paper chooses n-gram-based metrics, which are too strict and cannot capture the semantics between two sentences. For example, even if two sentences express a similar meaning, the different wording, paraphrases, and sequence length will result in a low score. This is reflected in Table 1, where the best model only gets an 8.2 BLEU score, meaning none of the models works here. In addition, those scores are not consistent, e.g., in Table 2, FTVC has no big difference with baselines in some metrics, but will suddenly be much higher/lower on some metrics. For captioning tasks, evaluation is challenging. Only relying on those automatic metrics will finally lead to a biased model that overfits the output style of a certain dataset. I suggest doing some human evaluation or using an LLM as the judge, which can be a more robust metric to quantify models' performance.

2. **Lack of Ablations and Analyses.** This paper only shows the main results on the evaluation datasets without ablating different components of the system. There are a lot of interesting analyses to do in the data generation and model training parts, e.g., will the number of generated image sequences affect the final performance? Why use Eq (2) to decide on the manipulation?

3. **Missing Baselines.** Adding the recent MLLMs (like LLAVA and QwenVL) as baselines will provide context on those benchmarks. In addition, fine-tuning those open-source MLLMs on your created METS datasets could help to disentangle the impact of data and modeling.

**Questions:**

1. Why is GPT-4V not evaluated on all other tasks?

2. Why not add METS when training on CLEVER and SPOT-THE-DIFF?

---

> ### Author Response · Authors · 2024-11-27
>
> We thank the reviewer for their feedback, the recognition of the technical soundness of our work, and the suggestions for improving the evaluation and analysis of our method. We address the reviewer's concerns as follows:
>
> ## Weaknesses
> 1. **Evaluation Metrics** We agree that n-gram-based metrics are not an ideal way to capture the semantics between two sentences.
> However, this is a common practice in the field of image captioning, and we follow the standard evaluation protocols to be able
> to compare our method with the existing methods, all of which report scores using these metrics.
>
> We have included an LLM based evaluation of the methods in Table 1. Please refer to the comment above for details: https://openreview.net/forum?id=rZxwa8JkJW&noteId=ZDpzdoct63
>
> 2. **Lack of Ablations and Analyses** We agree that there are many interesting avenues that perhaps could be explored in the future works.
> Unfortunately, data generation (and labelling) and model training are both computationally and monetarily expensive, and we were unable to
> explore all the possible ablations and analyses. We will include a discussion on the potential impact of the number of generated image sequences
> on the final performance in the revised manuscript.
>
> 3. **Missing Baselines** We agree that adding more baselines would provide a better context, however
> it is not trivial to make LLAVA/QwenVL work with multiple images, interleaved with text. Unlike VQA, image difference captioning
> is not a part of the standard LLM benchmarks, which was one of the motivations for creating the METS dataset. We will include an overview
> of more recent MLLMs in the revised manuscript.
>
> ## Questions
>
> 1. Running inference on GPT-4V costs money and we do not think it adds much value to the paper, since we are already comparing to SOTA methods for other tasks.
>
> 2. Both CLEVR Change and Spot-the-Diff datasets differ greatly from METS
> in terms of visual content, manipulation types and captioning style.
> Therefore, including METS in the training would not be beneficial.
> On the other hand, since METS is constructed to closely resemble real-world
> image editing scenarios, including it in the PSBattles training makes sense as
> it pushes the domain of the model from purely synthetic IP2P data to more realistic
> images of METS. Both CLEVR Change and Spot-the-Diff are already trained on in-distribution
> data, so adding METS would only add noise to the training process.

---

### Official Review · Reviewer_tBy4 · 2024-11-01

**Soundness:** 3
**Presentation:** 2
**Contribution:** 2
**Rating:** 5
**Confidence:** 4

**Summary:**

The paper presents Fusing Visual and Textual Cues (FVTC), a method for image difference captioning that leverages multiple sequential inputs. The method includes images and optional textual descriptions, to generate descriptive captions summarizing changes across multiple editing steps. The authors introduce the METS (Multiple Edits and Textual Summaries) dataset, comprising image sequences with detailed annotations and human-generated edit summaries. The study evaluates FVTC’s performance on established benchmarks and demonstrates superior results with multiple input images and textual data, outperforming baselines in captioning accuracy.

**Strengths:**

1. The FVTC model’s capability to incorporate multiple images and auxiliary text to improve difference captioning is novel and applicable to real-world scenarios where tracking sequential changes is crucial.
2. The new dataset is worth collecting. METS fills a gap by offering a dataset with longer, realistic editing sequences, supporting nuanced understanding of progressive edits.
3. FVTC is evaluated against multiple datasets with various metrics (e.g., BLEU, CIDEr, METEOR), showing robustness and adaptability across tasks.

**Weaknesses:**

1. The method depends on auxiliary textual data. While auxiliary data enhances model performance, it may limit FVTC’s application to cases where such data isn’t available, impacting generalizability. Please discuss or evaluate how FVTC performs without auxiliary textual data. Could these data be generated automatically in scenarios where the data is not readily available?

2. Concerns with task definition:

(1) The task of image difference captioning, as defined in this paper, focuses on scenarios where two images have nearly identical backgrounds with slight variations. However, this scenario is relatively limited in practical real-world applications, where more diverse changes are often encountered. For example, there are many image pairs in the Visual Storytelling Dataset (VIST) with a slight difference and shots from different viewing angles. Could these pairs be used in this method?

(2) Using semantic maps to highlight changes directly would be more explicit and interpretable for identifying differences between two images. For example, performing the change detection first and then describing the part of the change. Please include a comparison with semantic map methods if feasible. Captioning, by contrast, involves a degree of semantic encoding that translates visual differences into linguistic descriptions, making it difficult to accurately assess the model's true ability to detect and interpret nuanced changes.

3. Concerns with methodology: The paper relies heavily on synthetic data for training, which raises questions about the model's adaptability to real-world distributions. The extensive use of synthetic data may cause the model to become overly specialized in handling synthetic image patterns, which could differ significantly from real-world image characteristics, potentially limiting its generalization and robustness in practical scenarios. The authors could include experiments or analysis of the model's performance on real-world datasets. For example, some pairs could be constructed from MSCOCO or LAION-2B datasets. The authors could also discuss strategies for mitigating potential overfitting to synthetic data patterns.

**Questions:**

Please address my concerns above.

---

> ### Author Response · Authors · 2024-11-27
>
> We thank the reviewer for their feedback and acknowledgment of the novelty and potential of our work as well as the importance of the METS dataset.
> We appreciate the reviewer's concerns and suggestions for improving the generalizability of our model and dataset.
> We address the reviewer's concerns as follows:
>
> 1. We would like to emphasize that whilst the proposed method benefits from auxiliary textual data, it does not require it.
> In Table 1, we evaluate FVTC both with and without auxiliary textual data. Model versions that make
> use of the textual data are denoted with a "T" in the model name (e.g., FVTC-4**T**). Please refer to
> the "text" column in Table 1, which is marked with yes/no to indicate whether the model uses auxiliary
> textual data. Additionally, all of the results in Table 2 are from models that do not use auxiliary textual data.
>
> 2. Task definition:
>
> (1) Although the task of image difference captioning (IDC) could be extended to more diverse scenarios,  our work focuses on the important and timely scenario of manipulation and/or tampering with an image e.g. for misinformation, rather than viewing angle changes. One use case is to combine visual and textual data in open (C2PA) metadata for image provenance to help describe such edits to consumers.  We will further emphasise this focus vs. more general IDC in the paper.
>
> (2) We agree that there are cases where semantic maps could provide more explicit and interpretable
> description of the changes. For example using SAM or similar to parse and compare images.  In this work we explicitly focus on the task of image difference captioning, which
> is admittedly more challenging due to the linguistic nuances, but allows for a more natural
> description, especially in the context of a long list of edits, where summarization/omission is necessary.
>
> 3. We acknowledge there is a risk to overfit on any well represented domain in training data. However please refer to Table 2 for the results on Spot-the-diff and PSBattles, both of which are real-world datasets.
>
> We are not using MSCOCO or LAION-2B for pair construction but do use the OpenImages dataset, containing real world data, for constructing the METS dataset.

---

### Author Response · Authors · 2024-11-27

We thank the reviewers for their time and insightful comments, questions and suggestions.

Overall, the reviewers have found the proposed approach of incorporating multiple images
and auxiliary text data for the task of image captioning to be interesting and novel.
The reviewers also recognize the value and significance of the introduced METS dataset.

We have addressed each reviewer's concerns in the individual responses below.
However, one common question that was raised by multiple reviewers was regarding the
evaluation metrics used to assess the performance of the proposed approach. The reviewers
have suggested adding an LLM-as-judge evaluation metric to better caption the semantic
similarities between the text pairs. We agree that this is a valuable suggestion and
have added this evaluation metric to Table 1 in the revised manuscript, duplicated here for
quick reference.

We used GPT4 to score text pairs' semantic similarity as either 'low', 'medium', or 'high'.
In the table we report the percentage of replies with 'medium' L (M) and 'highl' L (H) scores.

| Model | images | text | L (M) | L (H) |
|-------|--------|------|-------|-------|
| GPT3.5 | 0 | yes | 16.2 | 0.6 |
| GPT4-V | 2 | no | 22.2 | 2.6 |
| GPT4-V | 2 | yes | 19.7 | 0.9 |
| GPT4-V | 4 | no | 26.9 | 1.9 |
| GPT4-V | 4 | yes | 24.1 | 1.2 |
| FVTC-2 | 2 | no | 22.6 | 9.4 |
| FVTC-2T| 2 | yes | 24.3 | 11.0 |
| FVTC-4 | 4 | no | 27.1 | 9.6 |
| FVTC-4T| 4 | yes | **30.1** | **12.4**|

---

### Meta-Review · Area_Chair_kTi9 · 2024-12-22

**Metareview:**

The main contribution of the paper lies in extending the scope of image difference captioning (IDC) with a sequence of intermediate image edit stages and textual inputs. It introduces a new dataset and a model to perform this task. The task appears to be novel and sound and the reviewers appreciate a new dataset being introduced. On the other hand, all reviewers have concerns about the motivation behind this work and/or how evaluation is done. Another concern is that the size of the proposed dataset is quite small (~1000).

The decision boils down to how these weaknesses impact the degree of significance of the proposed extension as compared to the de-facto IDC. The AC thinks the paper would still benefit from improving the prose and evaluation to illustrate such significance and the paper is not ready for publication at this time.

**Additional Comments On Reviewer Discussion:**

The authors clarified the motivation and added LLM-as-a-judge metrics during the rebuttal period. In the end, 3 of the 4 reviewers remain on the rejection side.

---

### Decision · Program_Chairs · 2025-01-22

Reject